# Properties of Particle Boards Containing Polymer Waste

**DOI:** 10.3390/ma16134774

**Published:** 2023-07-01

**Authors:** Marcin Kuliński, Joanna Walkiewicz, Dorota Dukarska, Dorota Dziurka, Radosław Mirski

**Affiliations:** Department of Mechanical Wood Technology, Poznań University of Life Sciences, ul. Wojska Polskiego 28, 60-627 Poznań, Poland; marcin.kulinski@up.poznan.pl (M.K.); joanna.walkiewicz@up.poznan.pl (J.W.); dorota.dukarska@up.poznanan.pl (D.D.); dorota.dziurka@up.poznan.pl (D.D.)

**Keywords:** particle board, polymer waste, blades, mechanical properties

## Abstract

Nowadays, a significant increase in interest in renewable energy sources can be observed. Wind farms have been one of the solutions representing this trend for many years. One of the important elements of windmills is the blades. The data indicate that what to do with the blades after their use is a global problem, and so it is important to find a way to recycle them. Hence, this work aimed to use these blades in the production of wood-based materials. Two fractions of a fragmented blade were used for the tests: a small one and large one. Boards characterized by densities of 650 kg/m^3^ and 700 kg/m^3^ were produced, in which the assumed substitution of the wood material with a polymer was 20% or 40%. Mechanical properties such as bending strength (MOR), modulus of elasticity (MOE), and internal bond strength (IB) were investigated. The 2S65 variant achieved the highest static bending strength and a modulus of elasticity of 2625 N/mm^2^. The second best result was noted for the 4S65 variant, which was significantly different from the 2S65 variant. In the case of the variants with a density of 700 kg/m^3^, no significant differences were found and their results were significantly lower. Moreover, research on thickness swelling (TS) after 24 h of immersion and water absorption (WA) were also conducted. The obtained results indicate that the manufactured boards are characterized by good physical and mechanical properties.

## 1. Introduction

The commonly used term “Ecology” comes from the combination of Greek words *oikos* (house) and *logos* (science), and is a branch of the natural sciences that deals with, among others, the study of interactions between organisms and the environment in which they live. People have always had a huge impact on the natural environment [1]. Constantly expanding urbanization leads to irreversible changes in the proper functioning of the ecosystem [2]. One of the ways to limit improper development is so-called “sustainable development”, which aims not only to protect the environment but also to improve quality of life [3]. The use and application of new technologies, based on energy from renewable sources, can sustainably help in the development of these dedicated areas. The most common sources of renewable energy are: wind turbines [4,5], photovoltaic panels [6,7], water turbines [8,9,10], biomass [11,12], and geothermal energy [13,14].

Wind farms, due to their ability to produce electricity, are a very popular solution in most countries. In 2022, these farms generated approx. 17% of the total demand for electricity in Europe and Great Britain, which was 487 TWh. Such amounts mean that the European Union, to meet the requirements for the production of energy from renewable sources, should build farms producing around 31 GW annually by 2030. Unfortunately, the construction of new farms is not enough because, at the same time, the number of wind turbines is decreasing due to aging-induced deterioration. Wind turbines that account for a total capacity of 5.6 GW are expected to be decommissioned over the next 5 years (of which 2.4 GW will be completely removed) [15].

The main parts of a wind turbine are: the blades, rotor, gear box, generator, nacelle, and tower [16]. The blades are the biggest problem for utilization. This is due to the fact that the blades are made of polymers reinforced with fibrous materials, which are most often glass and carbon fibers [17,18,19,20]. The WindEurope report from 2020 [21] estimated that 14,000 blades will be withdrawn from use, which translates into 40,000–60,000 tons of material. The data indicate that this is a global problem, so it is important to find a way to recycle them. Most often, the blades are stored in the ground. This is an inefficient solution as the material and energy are not recovered. The recycling of wind turbine blades is possible in three ways: through mechanical, thermal, and chemical processes [22].

Among the listed methods, only the mechanical one does not require such large financial resources for processing. Unfortunately, it is not possible to fully recover the material mechanically; rather, the mechanical method will only change its form. Reports in the literature focusing on the recycling of wind turbine blades show that after proper preparation, they can be used as elements of playgrounds, bicycle shelters, bridges, or bus stops [23]. Such solutions are easy and cheap to implement, so they can be successfully applied in many countries. A more demanding process is to cut the blades into smaller pieces and then ground them for use as filler material. For example, this method was used in the production of cement. This solution allowed the Swiss cement group Holcim to reduce the CO_2_ footprint during production by up to 16% while maintaining good strength parameters [22].

Wind turbine blades and other structures such as boats, yachts, bicycles, rods, or plates are manufactured as glass, glass–carbon, or carbon composites reinforced with epoxy resin. Therefore, various types of composites are used for the production of other materials. For example, the blades can be used to make stairs.

In the case of cement, good effects allow the process to be extended to other industries; therefore, the idea of using them in the production of wood-based panels is especially promising. Ground wind turbine blades, as a high-density material, in combination with wood particles at the proper ratio, should allow for the production of particleboard with a wide range of applications. In scientific research, polymer waste is being used more and more often for the production of wood-based materials. Atoyebi et al. [24] reported that the boards where wood dust was replaced by plastic fibers demonstrated an increase in their mechanical properties. The increase in mechanical characteristics of boards where high-density polyethylene (HDPE) and other plastics were used was observed by Jayaraman et al. [25] as well. However, there is no information in the literature on how particles of blades can be used for the production of wood-based materials. The blade particles, by increasing the density of the produced board, can have a positive effect on its mechanical properties. This research attempts to develop a method of combining wood-based materials with ground wind turbine blades. An innovative product produced in this way could have a positive effect on the environment on a global scale by solving the problem of wind turbine blade recycling. Hence, the main purpose of this work was to use blades in the production of wood-based materials.

## 2. Materials and Methods

In this study, the manufacture of single-layer particle boards was decided upon, although the preliminary analysis of the quality of the prepared material in the form of fragmented blades indicated that this type of particle substitution should be used for the inner layer of three-layer boards. This is because the process of grinding windmill blades is preferably carried out to obtain particles with quite large linear dimensions, especially for length and width. As for the wooden particles, pine (*Pinus sylvestris* L.) particles intended for the production of the middle layer of a three-layer particleboard/furniture board were used.

Two fractions of fragmented blades were used: small (Figure 1a) and large (Figure 1b). The fragmented material was obtained from ANMET (Szprotawa, Poland). The material was sieved using a sieve with a mesh size of 0.5 × 0.5 mm to remove very fine and dusty fractions. Hereinafter, the material is referred to as polymer chips or windmill blade chips.

It was decided to produce boards with a density of 650 kg/m^3^ and 700 kg/m^3^, and to replace the wood particles with polymer particles at an amount of 20% and 40% (the dry weight of chips to the dry weight of ground blades). When added to the wood particles, the ground blades’ moisture content was 0.9 ± 0.2%. Wood particles with an initial moisture content of 5.7 ± 0.3% were dried to a moisture content of around 2.7%. Moisture content was measured by using the drying-weighing method. The mixture of wood–polymer particles was combined using the pneumatic application of glue. The gluing degree was assumed to be 8% (the dry weight of the adhesive to the dry weight of the charge) for boards with a density of 650 kg/m^3^ and 6% for boards with a density of 700 kg/m^3^ (the dry weight of the glue to the dry weight of the wood (for the 80% wood content)). The scheme of the board manufacturing process is shown in Figure 2.

Urea-formaldehyde (UF) resin, supplied by Silekol (Kędzierzyn-Koźle, Poland) and characterized by a solid content of 76.5%, was used as an adhesive. Ammonium nitrate (40%) was used as the hardener in the amount of 1.5% (% dry mass of hardener/dry mass of resin). No agents improving the hydrophobicity of the manufactured boards were added to the particle–polymer mixture. The boards were pressed under standard conditions for this type of material: a temperature of the heating plate of 180 °C, a pressing time of 30 s per mm of the assumed board thickness (thickness of manufactured boards was 15 mm), and a unit pressure of 2.2 MPa. After production, two samples were taken from the central part of the boards to evaluate the moisture content (MC). The MC of the boards ranged from 1.36% to 1.89%. Three boards were produced for each variant.

The following system to describe the XYZZ variants was adopted, i.e.:-X percentage of fragmented blades: 2 = 20%, 4 = 40%;-Y type of agglomerate/polymer chips: S = fine, B = large;-ZZ density of manufactured boards: 65 = 650 kg/m^3^, 70 = 700 kg/m^3^.

Thus, the description of the samples is given in Table 1.

The produced boards were conditioned for two weeks at 20 ± 2 °C and 60 ± 5% relative humidity. After the air-conditioning period, the MC of the boards from which test samples were cut was 5.8 ± 0.3%. The number of repetitions for each experiment was at least 12. The only exception was the density profile, where 3 samples were used.

After the board conditioning, the produced boards were tested in terms of the following parameters according to the relevant standards:-Bending strength (MOR) and modulus of elasticity (MOE), according to EN 310 [26];-Internal bond (IB) strength, according to EN 319 [27];-Thickness swelling (TS) after 24 h, according to EN 317 [28], and water absorption (WA).

The mechanical properties were assessed using a TinusOlsen 10 K testing machine. The machine, with a maximum allowable load of 10 kN, is fully automatic, and the software calculates the modulus of elasticity in the bending test. The density profile was determined using a Grecon Dax6000 profilometer (Fagus-GreCon Greten GmbH & Co. KG., Alfeld/Hanover, Germany). 

To illustrate the distribution of polymers in the structure of the boards, samples with dimensions of a 50 × 50 × thickness were analyzed using a tomograph. Scanning was performed with a Hyperion X9Pro CT scanner with a resolution of 0.15 mm at a tube voltage of 90 kV, a spot resolution of 68 m, and a maximum imaging field of 13 cm × 16 cm (MyRay, Via Bicocca, Imola, Italy). The Hounsfield scale (HU) was used. The Hounsfield unit is a relative quantitative measurement of radio-frequency density used by radiologists to interpret computed tomography (CT) images. Although it is not a very precise measure, it allows for the determination of changes in the wood-based material.

The obtained results were analyzed statistically and compared with the results of previous studies. Statistica software version 13.0 (Version 13.0, StatSoft Inc., Tulsa, OK, USA) was used for statistical analysis.

## 3. Results and Discussion

As a preliminary test, the strength of the adhesive joint was assessed to check whether the urea-formaldehyde resin allows for the sufficient bonding of polymer chips with wood chips. The assessment was made relatively simple, i.e., by determining the tensile strength perpendicular to the planes (Figure 3). At the same time, it was assumed that the joint would probably be damaged rather than being pieces of polymer or a piece of plywood. Pieces of blades with dimensions of 50 × 50 mm (the thickness as intended for grinding) were prepared for the tests. Three different variants were received from Anmet (Figure 4). The samples differ in both the thickness and quality of the material from which they were made.

Eight samples were prepared for each variant and the results were analyzed together. The obtained average tensile strength result was not impressive and was only 0.46 N/mm^2^ (Figure 5). However, it was also not low enough to abandon all further testing. All samples failed when assessing the adhesive joint. Quite a large variation in the tensile strength values of individual samples was observed (Figure 5). This may be due to the surface quality of the analyzed samples. In the case of cutting the material for testing directly from the blades of windmills, it is not easy to maintain the parallelism and smoothness of the surface, which influences the actual size of the surface involved in the gluing of the elements. As seen in the middle sample (Figure 3), the area covered by the adhesive is slightly smaller than the sample area.

The basic parameter that allows for the quick quality control of the technological line and the expected mechanical properties of the produced boards is the density profile. Its course allows us to determine the efficiency of the forming equipment and estimate the quality of the produced boards. Since the density of the implemented polymer particles was much higher than in the case of wood, being up to three times that in the case of pine wood and even exceeding 2000 kg/m^3^, they caused a significant distortion in the density profile. As shown in Figure 6, the tested samples were not characterized by a density profile similar to the profiles of wooden materials. Areas of very high density appeared, regardless of the distance from the sample surface. The application of finer polymer particles resulted in smaller density changes within the cross-section of the sample. It was also possible to observe some displacement or migration of the polymer particles towards one plane (the forming plane). Therefore, the analysis of such a course is difficult. However, it will probably be possible to determine the expected mechanical properties based on an analysis of many samples. As mentioned earlier, this type of material should be used to substitute for particles in the middle layer of three-layer boards. In this case, the observed changes would be transferred to deeper layers and would not have such a significant impact.

Figure 7 presents the results of the analysis of variance (ANOVA) of the outcomes of static bending strength of the manufactured wood–polymer boards. This is a very simplified approach because not only did the boards differ in density and the degree of gluing, but also in the quantity and dimensional quality of the polymer particles.

It was observed that large polymer particles did not produce as good of results as the finer chips. The differences in static bending strength between the S70 and B70 boards were very clear, reaching about 25%, and these differences were statistically significant. Especially in the case of higher density boards, there was no significant effect of the share of polymer chips on the board structure. Therefore, it can be assumed that they behave similarly to wood chips. The lower strength of boards made of polymer particles with larger linear dimensions may result from two factors. Firstly, because they have a higher density than wood and larger linear sizes, their gluing is less efficient when carried out in the available slow-speed gluing machines. Secondly, their distribution within the board structure is worse than that of fine particles, which is confirmed by the density profiles. The relatively low static bending strength of the produced boards should not raise any concerns, especially for boards with a density of 700 kg/m^3^, because in their case the real gluing degree was only 4.8%. Such a low gluing degree resulted from the fact that the manufactured boards were to be referred to as the middle layer for three-layer boards. The authors’ experiment shows that the strength of such boards should be higher than 6 N/mm^2^. The produced boards were characterized by a high modulus of elasticity, ranging from over 1500 N/mm^2^ to nearly 2630 N/mm^2^ (Figure 8). Although this requirement is not imposed, boards of this type should have an average modulus of elasticity of 1600 N/mm^2^. Therefore, this condition will be met by all boards made with fine polymer chips.

As in the case of static bending strength, there was no clear effect of the share of polymer particles on the structure of boards with a density of 700 kg/m^3^. Statistically significant differences occurred between 2S65 and 4S65 boards, i.e., boards made of fine chips with a lower density but with a higher degree of gluing. In their case, a larger share of polymer particles significantly reduced the modulus of elasticity. Since a higher modulus of elasticity characterizes the systems of carbon or glass fibers in combination with epoxy resin compared to wood, the observed differences in the modulus of elasticity may indicate poorer anchoring of the polymer particles in the structure of the boards. A certain confirmation of this is the lower tensile strength perpendicular to the planes of the 4S65 boards as compared to that of the 2S65 boards, i.e., with a lower proportion of polymer particles. What is also surprising is that the strength of all manufactured boards was similar. The lower degree of gluing of boards with a density of 700 kg/m^3^ was compensated for by their higher density. However, such a high tensile strength was not expected. The fact that it can be higher for boards than the demonstrated bondability of the polymers used was to be expected. With such a proportion of polymer particle to pine particle, it is the pine particle that may determine the tensile strength.

The relatively high density in the inner part of the board, being close to 600 kg/m^3^, is probably why the analyzed boards achieved such high tensile strength values perpendicular to the planes. In the case of furniture particleboards, the density in the center of the board may be lower than 500 kg/m^3^ and may oscillate around 460–480 kg/m^3^. Although the ANOVA did not show significant differences between individual boards regarding the tensile strength perpendicular to the planes, one more analysis was performed for the S65 boards (Figure 9). Only these two variants of the boards were compared with each other using Student’s *t*-test. The analysis showed that the strength of these two types of boards differed statically (*p* = 0.013356). Therefore, the lower modulus of elasticity in the case of these boards may be caused by the quality of the gluing of individual chips. Figure 10 shows differences in the appearance of the board surface resulting from the amount of introduced polymer particles and their quality. The finer polymer chips blend better into the structure of the manufactured polymer–particle boards.

The fact that the 2S65 boards were better than the 4S65 boards is also evidenced by their resistance to water (Figure 11). Boards with a smaller amount of hydrophobic material show lower thickness swelling and lower water absorption than boards with twice as much of a share of polymer chips. In general, such results of swelling and water absorption can be considered to be low. Laboratory single-layer particleboards made of only pine chips showed 30% swelling and a water absorption close to 90%. In the case of S65-type boards, with a degree of gluing similar to commonly produced ones, swelling below 20% can be considered more favorable (than what only results from introducing non-water-absorbing material into the boards). In the case of type 70 boards, the expected decrease in swelling and water absorption can be noted with the increase in the proportion of polymer particles. However, no clear influence of the type of introduced polymer particles was observed.

With the use of computed tomography, it is possible to observe, e.g., how the polymer particles are arranged in the board’s structure, and to make other interesting observations. As shown in Figure 12, the distribution of polymer particles in the analyzed fragment of sample 2S70 was relatively even. Both wood and polymer particles do not show any clear orientation as well. In the case of sample 2B70, there were some clusters of polymer particles in the analyzed area. However, their clear arrangement in the direction of one of the axes cannot be observed either.

During the analysis of the images, areas of a much higher radiation density than expected were observed in the tested samples (Figure 13). The radiation density of the polymer particles was about 2500 HU. The observed value of more than 14,700 HU is therefore much higher than that assigned for polymer particles. Such high HU values correspond to steel. Steel particles were found in both wood particles and polymer particles. However, they are more likely to come from the fragmented windmill blades.

Computed tomography shows the distribution of polymer particles in the board structure much better than the density profiles. However, as the tests of the mechanical properties of the produced boards showed, the polymer particles obtained from the grinding of windmill blades worked well with the wood particles and urea-formaldehyde resin. Therefore, comprehensive research on the relationship between mechanical properties and the density profile for three-layer boards will show a simple relationship that allows for a quick analysis of production processes. However, despite the significantly different densities (about 550 kg/m^3^ vs. 1950 kg/m^3^ for pine and polymers, respectively), the polymer chips’ behavior should be considered very favorable.

## 4. Conclusions

The tests have shown that using fragmented blades of power windmills does not cause major problems when particles/fragments of the ground blades are introduced into the middle layer of boards. The obtained test results for the analyzed fractions, their amount, or the amount of glue indicate that good physical and mechanical properties should characterize the resulting boards.

The 2S65 variant achieved the highest static bending strength and modulus of elasticity of 2625 N/mm^2^. The second best result was the 4S65 variant, which was significantly different compared to the 2S65 variant. In the case of variants with a density of 700 kg/m^3^, no significant differences were found and their results were significantly lower.

The authors’ experience and simple calculations show that single-layer boards should have a strength of at least 6 MPa, and thus they can be the center layer when producing three-layer boards.

Therefore, all manufactured variants allowed us to achieve the expected levels of bending strength and modulus of elasticity during the industrial production of three-layer boards.

## Figures and Tables

**Figure 1 materials-16-04774-f001:**
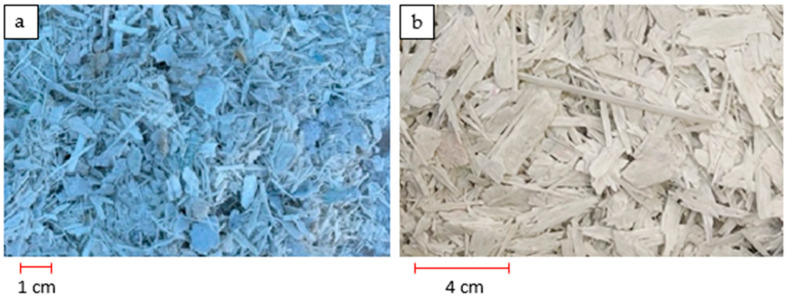
Polymer particles representing: (**a**) fine fraction; (**b**) large fraction.

**Figure 2 materials-16-04774-f002:**
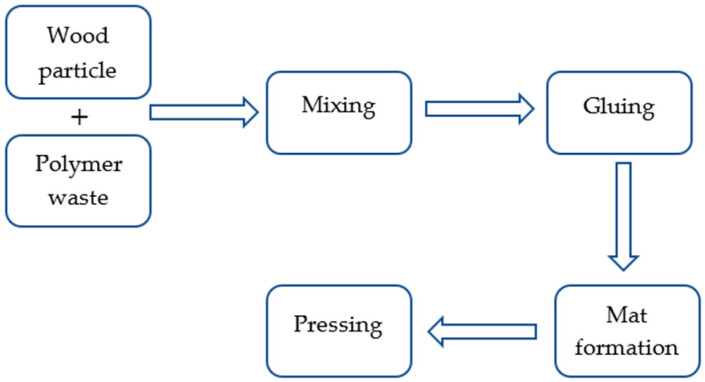
The scheme of the board manufacturing process.

**Figure 3 materials-16-04774-f003:**
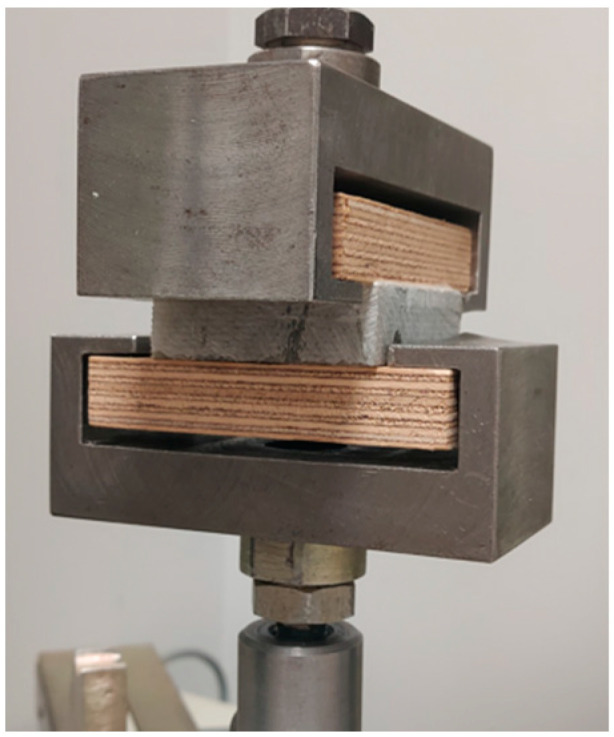
Sample in the machine holder.

**Figure 4 materials-16-04774-f004:**
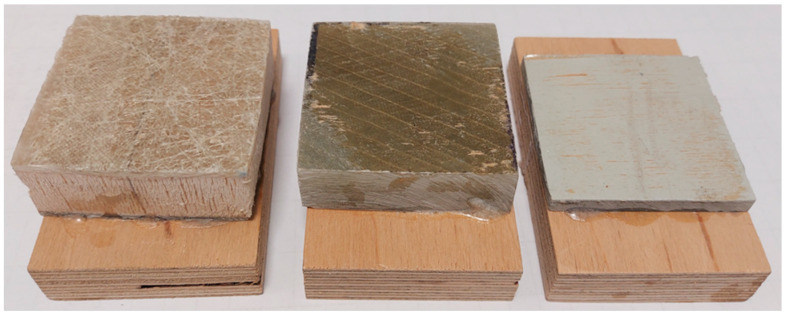
Examples of analyzed variants of fragmented windmill blades. From left: reinforcing rib in the middle of the propeller (between the composite balsa wood layers), the surface of the propeller closer to the hub, and the surface of the propeller at its end.

**Figure 5 materials-16-04774-f005:**
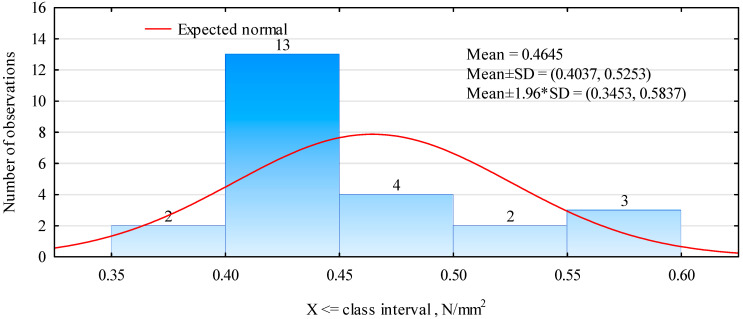
Histogram of tensile strength perpendicular to the planes of the plate (polymer cubes).

**Figure 6 materials-16-04774-f006:**
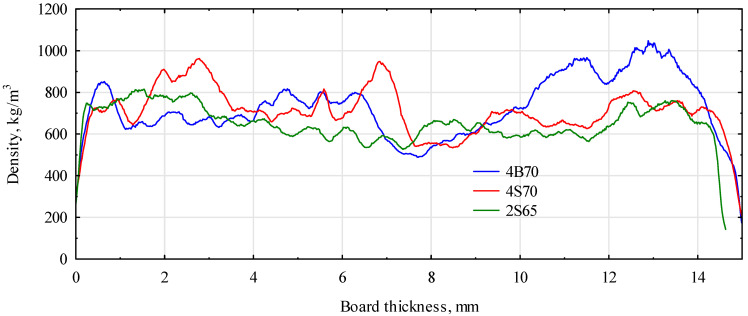
Selected density profiles for the analyzed particle–polymer boards.

**Figure 7 materials-16-04774-f007:**
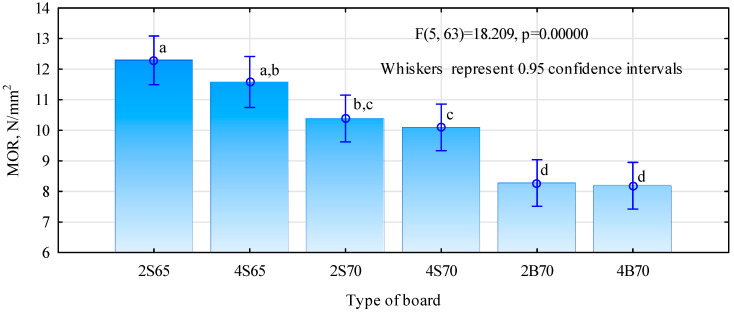
Static bending strength of manufactured polymer chip boards. Homogeneous groups are marked with lowercase letters.

**Figure 8 materials-16-04774-f008:**
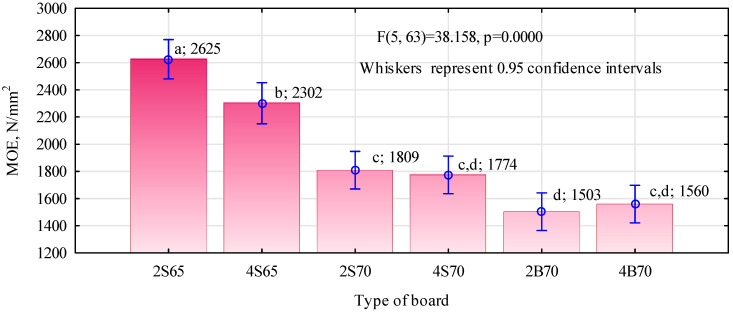
Static bending strength of manufactured wood–polymer boards. Homogeneous groups are marked with lowercase letters.

**Figure 9 materials-16-04774-f009:**
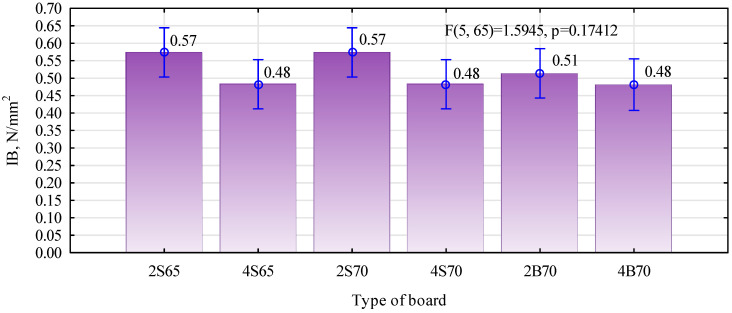
Tensile strength is perpendicular to the planes of the manufactured particle–polymer boards.

**Figure 10 materials-16-04774-f010:**
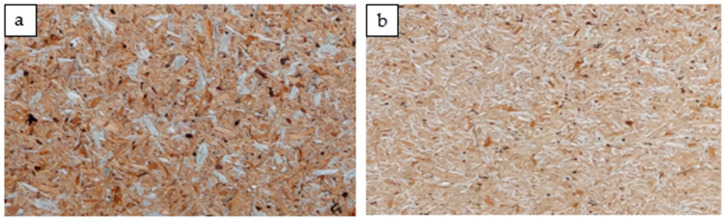
The appearance of the boards after pressing: (**a**) 2B70; (**b**) 4S65.

**Figure 11 materials-16-04774-f011:**
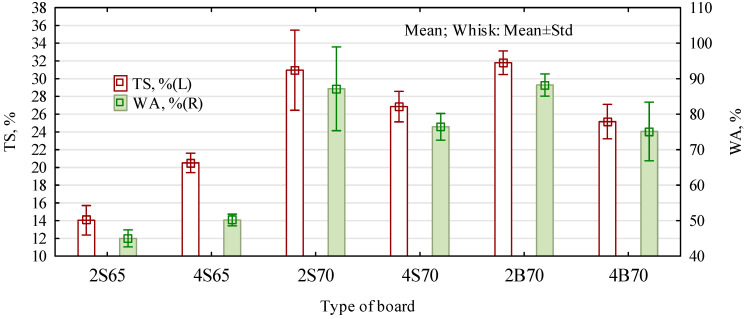
Resistance to water of the manufactured particle–polymer boards.

**Figure 12 materials-16-04774-f012:**
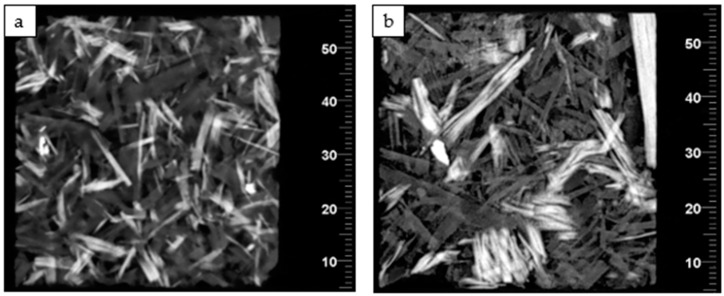
Tomographs showing the distribution of polymer chips in the board structure of (**a**) 2S70 sample; (**b**) 2B70 sample.

**Figure 13 materials-16-04774-f013:**
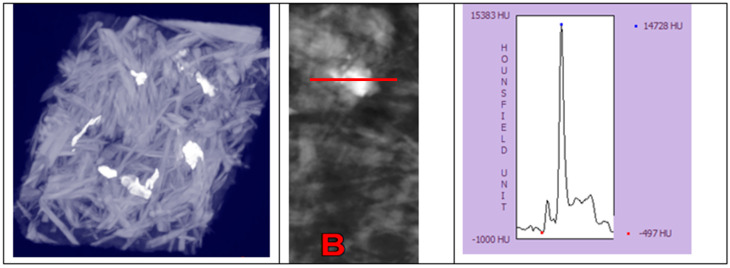
Image of the 4S70 sample with an analysis of the radiation density distribution passing through the clearly bright area seen in the middle image (scale about 1:1). Photo B shows the measurement site.

**Table 1 materials-16-04774-t001:** Description of variants.

XYZZ	“X”—Percentage of Fragmented Blades	“Y”—Fraction of Polymer Particles	“ZZ”—Density of Manufactured Boards
2S65	20%	Fine	650 kg/m^3^
4S65	40%	Fine	650 kg/m^3^
2S70	20%	Fine	700 kg/m^3^
4S70	40%	Fine	700 kg/m^3^
2B70	20%	Large	700 kg/m^3^
4B70	40%	Large	700 kg/m^3^

## Data Availability

The data presented in this study are available upon request from the corresponding author.

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
