# Peer review of "Properties of Particle Boards Containing Polymer Waste"

_materials, 2023, doi:10.3390/ma16134774_

Round 1
Reviewer 1 Report
The manuscript shows an idea of combining polymer from turbine blades with wood to produce wood-based panels. To me, it needs a Major revision. Please kindly clarify the following matters when revising.
1. The Intro is more focused on the polymer part of turbine blades. However, some focus should be shifted towards the necessity of wood-based panels sourced from polymer waste.
2. How the amount of Urea-formaldehyde (UF) resin and Ammonium nitrate was selected? Any optimisation? These two can have a significant impact on the properties.
3. Authors should provide more explanation on "despite the significantly different densities, the polymer chips' behaviour should be considered very favourable"
4. "It was decided to produce boards with a density of 650 kg/m3 and 700 kg/m3 , and replace wood particles with polymer chips of 20% and 40%" Please explain the basis upon which the decision was made, any optimisation or literature support?
5. The manuscript describes satisfactorily its results. However, how these findings are compared with current literature or at least towards the materials for intended applications? There is probably no reference in the discussions. This part needs a detailed and deeper discussion.
6. It would be better to provide a Table in Methodology to identify the sample names more clearly.
7. Conclusions should be improved by adding more information on the key findings.
Minor spell errors are present, such as line 48: utylisation.
Author Response
Dear Reviewer,
thank you very much for your contribution to our development.
The manuscript shows an idea of combining polymer from turbine blades with wood to produce wood-based panels. To me, it needs a Major revision. Please kindly clarify the following matters when revising.
- The Intro is more focused on the polymer part of turbine blades. However, some focus should be shifted towards the necessity of wood-based panels sourced from polymer waste.
Thank you for your suggestion. The information has been added.
- How the amount of Urea-formaldehyde (UF) resin and Ammonium nitrate was selected? Any optimisation? These two can have a significant impact on the properties.
The content of hardener and resin was selected based on the authors' experience.
- Authors should provide more explanation on "despite the significantly different densities, the polymer chips' behaviour should be considered very favourable"
The density of pine wood is about 550 kg/m3, and that of polymer chips/particles is about 1900-2000 kg/m3. Therefore, polymer chips are 4 times heavier than pine chips. For these reasons, they behave differently when gluing and forming the carpet.
- "It was decided to produce boards with a density of 650 kg/m3 and 700 kg/m3 , and replace wood particles with polymer chips of 20% and 40%" Please explain the basis upon which the decision was made, any optimisation or literature support?
Boards with a density of 650 kg/m3 and 700 kg/m3 are commonly produced and used in construction. Therefore, it was decided to use such densities, while the share of 20 and 40% was aimed at using as much polymer material as possible while preserving a larger part of the wood material. In addition, two contributions were made to verify whether the contribution had an effect on the board properties.
- The manuscript describes satisfactorily its results. However, how these findings are compared with current literature or at least towards the materials for intended applications? There is probably no reference in the discussions. This part needs a detailed and deeper discussion.
The article did not use a comparison with literature data, because no data was found on a similar product using this or a similar material (windmills). However, the tests were carried out in accordance with the standards for wood materials.
- It would be better to provide a Table in Methodology to identify the sample names more clearly.
Thank you for your suggestion, the table has been added.
- Conclusions should be improved by adding more information on the key findings.
Thank you for your suggestion, more information about results were added.
Reviewer 2 Report
The article contains a detailed scientific research. This study aims to create a technique for mixing in shredded wind turbine blades with wood-based components. It is absolutely necessary to make some updates that I have mentioned below before publication.
1. The magnifications of the images taken for Figures 1, 11 and 12 should be indicated below the figure. In addition, it should be stated with which device or method these images were taken.
2. It will be more understandable if the method section is titled separately in a way to describe each study separately.
3. It is necessary to eliminate the resolution problem in the figures.
4. I think that the number of references used in the article is low. Especially at the stage of interpretation of the results, it would be more appropriate to make a comparison by making waste to the reference.
Minor editing of English language required
Author Response
Dear Reviewer,
thank you very much for your contribution to our development.
The article contains a detailed scientific research. This study aims to create a technique for mixing in shredded wind turbine blades with wood-based components. It is absolutely necessary to make some updates that I have mentioned below before publication.
- The magnifications of the images taken for Figures 1, 11 and 12 should be indicated below the figure. In addition, it should be stated with which device or method these images were taken.
The photograph for figure 1 was taken with a mobile camera. The pictures for Figures 11 and 12 (12 and 13) were taken using a CT scanner. The specifications of the tomograph are given in the methodology. Scales have been added for Fig. 1. Tomographs are approximately 1:1 scale
- It will be more understandable if the method section is titled separately in a way to describe each study separately.
Thank you for your suggestion, however, in the opinion of the authors, the methodology is described clearly.
3.It is necessary to eliminate the resolution problem in the figures.
We have no influence on the quality. Fig. 12 is illustrative, while Fig. 13 can be removed.
- I think that the number of references used in the article is low. Especially at the stage of interpretation of the results, it would be more appropriate to make a comparison by making waste to the reference.
Thank you for your suggestions, this is valuable feedback. However, at present it is difficult for us to find suitable publications. Perhaps the reviewer has more experience in this topic. On the other hand, references to unrelated publications may be poorly received by other readers
Reviewer 3 Report
Dear Authors
The manuscript is interesting. It is about solving one of environment problem in case of recycling residues of windfarms structure such as windmill blades.
Please consider following major comments:
1-Add some important obtained numerical results of your study in abstract and conclusion sections.
2-In line 49, you write that blades were constructed by reinforced polymers. Please write main polymers utilized for this purpose, obviously.
3-As you used windmill blades, please show various layers and materials utilized in construction of it. Please explain with a schematic picture. It helps readers to better underrating. Which kind of polymer and other materials were used in your selected blades?
3-Which kind of wood was used? Please describe about source of it. Is it a recycle wood or original?
Why did you select this type of wood?
What were specifications of used wood particles such as thickness and particle size? Please describe about it in text.
4-Please explain board construction steps schematically.
5-Which method used for determining water content of various samples?
6-In line 153, you write various samples were used. Please describe them, completely in this section. Add results of analyzing every one sample.
7-In Fig. 4, It is not clear which column is related to which sample. Please explain clearly.
8-In Figs. 6, 7, and 8, It is not clear that the results were obtained from which samples. Please explain clearly.
9-What reasons and sources were resulted to water absorption? Did you try to decrease water absorption and consequently to enhance final samples functionality?
10-What are applications of your prepared samples?
In general, your study is interesting, but it was not well-written and the results were not presented clearly.
11-There are numerous spelling and grammatical errors that must be corrected.
Sincerely
There are numerous spelling and grammatical errors that must be corrected.
Author Response
Dear Reviewer,
thank you very much for your contribution to our development.
The manuscript is interesting. It is about solving one of environment problem in case of recycling residues of windfarms structure such as windmill blades.
Please consider following major comments:
1-Add some important obtained numerical results of your study in abstract and conclusion sections.
Thank you for your suggestion, important results were added to abstract and conclusion.
2-In line 49, you write that blades were constructed by reinforced polymers. Please write main polymers utilized for this purpose, obviously.
The most common are glass and carbon fibers. Intensive work is currently underway to recover these fibres. Such a fiber is a recycled fiber and can be partly recycled in production.
3-As you used windmill blades, please show various layers and materials utilized in construction of it. Please explain with a schematic picture. It helps readers to better underrating. Which kind of polymer and other materials were used in your selected blades?
Thank you for your suggestion. But we do not have access to information on what polymers have been used in the blades we use.
4-Which kind of wood was used? Please describe about source of it. Is it a recycle wood or original? Why did you select this type of wood? What were specifications of used wood particles such as thickness and particle size? Please describe about it in text.
Line 88 states that it is pine wood. The selected material is commonly used in the production of wood-based materials and parameters such as particle size, dimensions are widely known.
4-Please explain board construction steps schematically.
A schematic graph has been added to the article
5-Which method used for determining water content of various samples?
The information has been added to the article.
6-In line 153, you write various samples were used. Please describe them, completely in this section. Add results of analyzing every one sample.
And
7-In Fig. 4, It is not clear which column is related to which sample. Please explain clearly.
The purpose of this study was to determine the adhesion of the adhesive to ground windmill blades. The type of sample does not matter, because all of them were destroyed in the adhesive joint. Fig. 7 shows a histogram of the obtained results. Since the tensile strength of such polymers is very high, the samples were not differentiated. Approximately 8 samples were prepared and tested randomly.
8-In Figs. 6, 7, and 8, It is not clear that the results were obtained from which samples. Please explain clearly.
Descriptions are added to the drawings, and references to specific samples are also made in the text. We do not fully understand the reviewer's comment.
9-What reasons and sources were resulted to water absorption? Did you try to decrease water absorption and consequently to enhance final samples functionality?
These are preliminary tests and no attempt was made to reduce the water absorption of the finished board at this stage. Due to the fact that it is a porous material, mostly made of wood, the boards can absorb water.
10-What are applications of your prepared samples?
We believe that wind turbine blades can be a good substitute for wood chips. Due to their properties, they can be introduced into chipboards, mainly into the middle layer.
In general, your study is interesting, but it was not well-written and the results were not presented clearly.
11-There are numerous spelling and grammatical errors that must be corrected.
Thank you for your attention, the article has been corrected.
Round 2
Reviewer 1 Report
The authors have addressed my comments adequately.
Reviewer 3 Report
Dear authors
The manuscript can be published.
Sincerely
It is OK.